# Heterosis and Differential DNA Methylation in Soybean Hybrids and Their Parental Lines

**DOI:** 10.3390/plants11091136

**Published:** 2022-04-22

**Authors:** Liangyu Chen, Yanyu Zhu, Xiaobo Ren, Dan Yao, Yang Song, Sujie Fan, Xueying Li, Zhuo Zhang, Songnan Yang, Jian Zhang, Jun Zhang

**Affiliations:** 1Faculty of Agronomy, Jilin Agricultural University, Changchun 130118, China; 20200045@mails.jlau.edu.cn (L.C.); zhuyyjlau@163.com (Y.Z.); rxb424571572@163.com (X.R.); songyangjlnd@163.com (Y.S.); fansujie@jlau.edu.cn (S.F.); xueyingl@jlau.edu.cn (X.L.); zhuoz@jlau.edu.cn (Z.Z.); 2College of Life Science, Jilin Agricultural University, Changchun 130118, China; dyao@jlau.edu.cn; 3National Crop Variety Approval and Characteristic Identification Station, Jilin Agricultural University, Changchun 130118, China; 4Department Biology, University of British Columbia, Okanagan, Kelowna, BC V1V 1V7, Canada

**Keywords:** soybean, heterosis initiation and formation, DNA methylation, reciprocally hybrid seeds

## Abstract

Heterosis is an important biological phenomenon and is widely applied to increase agricultural productivity. However, the underlying molecular mechanisms of heterosis are still unclear. Here we constructed three combinations of reciprocal hybrids of soybean, and subsequently used MethylRAD-seq to detect CCGG and CCWGG (W = A or T) methylation in the whole genome of these hybrids and their parents at the middle development period of contemporary seed. We were able to prove that changes in DNA methylation patterns occurred in immature hybrid seeds and the parental variation was to some degree responsible for differential expression between the reciprocal hybrids. Non-additive differential methylation sites (DMSs) were also identified in large numbers in hybrids. Interestingly, most of these DMSs were hyper-methylated and were more concentrated in gene regions than the natural distribution of the methylated sites. Further analysis of the non-additive DMSs located in gene regions exhibited their participation in various biological processes, especially those related to transcriptional regulation and hormonal function. These results revealed DNA methylation reprogramming pattern in the hybrid soybean, which is associated with phenotypic variation and heterosis initiation.

## 1. Introduction

Heterosis or hybrid vigor refers to the better performance of hybrid offspring over their parents. In plants, heterosis can be observed in traits such as growth, yield and environmental adaptation [1]. Genetic explanations for heterosis include three possible mechanisms [2]: dominance, overdominance and epistasis. The theoretical basis of heterosis derives from the global heterogeneity of the parental genomes [3,4], but some reports suggested that an individual gene can cause the hybrids to show phenotypic advantage as well [5,6,7]. The molecular basis of heterosis remains largely unknown, which hinders its further use in plant production. 

In recent years, omics studies have revealed that broad changes in gene activity between hybrids and their parents cause gene expression complementation and thus lead to hybrid vigor [1]. These changes affect a variety of biological processes, including energy utilization and metabolism, light and hormone signaling, stress responses and ageing [8]. Accumulating evidence also implicates epigenetic variation in heterosis, by regulating gene expression and stabilizing the genome [9]. DNA methylation, as an epigenetic modification, occurs at cytosine within all sequence contexts, including CG, CHG and CHH contexts (where H = A, T or C). Several studies have shown that hybrids of many crops undergo rearrangements in DNA methylation causing their methylome to vary from their parental lines, such as rice [10], maize [11] and pigeonpea [12]. Furthermore, the loss of genome methylation of transposable elements (TE) in hybrids led to a loss of phenotypic superiority [13,14]. Work in rice indicates that a role for parental CHG methylation divergence in allelic-specific expression is believed to be associated with hybrid vigor [15]. Some heterosis-related genes showed strong correlation between expression levels and methylation in pigeon pea [12]. However, dissecting the role of DNA methylation in heterosis is complex because genome-wide DNA methylation profiles varies according to developmental stages and environmental factors. Studies using *Arabidopsis* and pigeon pea seedlings showed increased DNA methylated levels in reciprocal hybrids [12,13]. Rice developing seeds showed that overall DNA methylation of reciprocal hybrids declined in contrast to those of parents [16]. In 6-week-old rice seedling leaf or root, changing DNA methylation in hybrids did not cause appreciable alterations for the target genes expression [17]. In maize, expanded leaves at the 7–8 leaf stage also showed no correlation with grain-yield heterosis detected for parental differences in CG or total (CG + CHG) methylation levels [18]. 

The heterosis of a trait may be the result of the accumulation of several phenotypic advantages in different developmental stages, meanwhile it is initiated as early as double fertilization, such as maize hybrid embryos at 6 days of fertilization having a phenotypic dominance [19]. Embryos and endosperm synchronously undergo dramatic changes in the DNA methylome when reciprocal hybrids and their parents are compared, probably causing the heterosis of immature seed-related traits [16]. In particular, DNA methylation is differentially altered in diverse cytosine contexts during soybean seed development: CHH increases significantly during development throughout the entire seed and drops precipitously within the germinating seedling, while only marginal changes in the global level of CG/CHG occur during the same developmental period [20,21]. Moreover, a loss of CHH and CHG methylation does not affect seed development in *Arabidopsis*, and the key genes of soybean seed development are present in genomic regions that are not methylated [21]. The above information suggests that methylation profiles formed during seed development may have other functions. On the other hand, hybridization can induce changes in DNA methylation in hybrids, which are maintained in the hybrid progenies [22]. In addition, epigenetic modifications formed during seed development could be transmitted via cell division to exert their function after a certain growth stage [23]. Hence, DNA methylome changes during seed development of contemporary hybrid seeds may lead to “strong” seed laying the material basis for hybrid advantage of other traits, or potential transmission to subsequent developmental stages responsible for the initiation of hybrid dominance.

Soybean [*Glycine max* (L.) Merr.] is one of the most important oilseed crops in the world. Insufficient soybean production in China and many developing countries is a big challenge; yield of soybean needs to be improved rapidly [24]. Existing studies have demonstrated significant yield of soybean hybrids over their parents [25,26,27,28], but our knowledge of soybean heterosis is very superficial; in addition, the effects of epigenetic mechanisms have rarely been reported. In this work, we investigated DNA methylation differences between three reciprocal hybrids and their parental lines in the middle developmental period of contemporary seed by MethylRAD-seq (methylation-dependent restriction-site associated DNA sequencing) that can identify CCGG and CCWGG (where W = A or T) methylated sites in the whole genome [29]. Our present results would provide better understanding and help to elucidate the initiation, formation and regulatory mechanisms of heterosis.

## 2. Results

### 2.1. Genome-Wide Profiles of CCGG and CCWGG Modifications in Developing Seeds of Reciprocal Hybrids and Their Parents

To profile CCGG and CCWGG methylation states in developing seeds of hybrids as compared to their parents, we mapped MethylRAD-Seq reads to the soybean reference genome Wm82.a2.v1. Only uniquely mapped reads were used for further analysis. After alignment, 49,967 CCGG sites and 85,999 CCWGG sites were obtained, respectively. Subsequently, we calculated the average methylation levels (reads per million/RPM) for each sample and for different regions of the genome within each sample. Results showed that the methylation levels of CCWGG sites were significantly less methylated than CCGG sites in all samples (Figure 1A,B and Appendix A). The most significant reduction of methylation was detected at transposable elements (TE) (Figure 1A,B and Appendix A). Further analysis of methylation patterns of parents and reciprocal hybrids revealed that levels of CCGG methylation were roughly similar in all examined samples, while CCWGG methylation was significantly different between P3 and its crossing offspring (F13 and F31) and between F32 and its parents or F23 (Appendix A). The mean methylation levels of all hybrids were slightly lower than the corresponding mid-parent value (MPV), except for the levels of F32 at CCWGG sites, and the reason for this was that the methylation levels in whole genome regions at these sites of F32 were higher than the MPV (Figure 1A and Appendix A). These results indicated that remodeling of the methylation profile occurred in F_1_ hybrids and slight decrease in total levels occurred compared to MPV.

Comparisons of the distribution trends of the two epigenetic modified sites in the soybean genome showed that CCGG sites have higher distribution levels among the gene region (including the gene body and its upstream and downstream 2000 bp) than CCWGG sites, while more CCWGG sites were in the intergenic region (Figure 1C). Furthermore, there were significant differences in average methylation levels in different regions of the genome, with the levels significantly higher at TE and intergenic regions than in other regions, regardless of cytosine context. For CCGG sites, methylation levels of gene bodies were marginally lower than regions 2000 bp upstream or downstream. In contrast, gene body methylation of CCWGG sites was significantly higher than in the upstream or downstream flanking region (Figure 1A and Appendix A). We also found that many hyper-methylated sites were present in approximately equal numbers in each sample (methylation level of these sites is at least five times average methylation level of the same cytosine context) at two cytosine contexts, and the majority of these sites are located in the TE and intergenic regions (Appendix A and Appendix A). 

### 2.2. Partial Differential Methylation Sites in Reciprocal Hybrids Are Due to Selective Inheritance of Parental Methylation Status

To compare DNA methylation of all hybrids with their parents, two methylated sites of parental lines were first divided into four categories in terms of methylation levels, including those for which one parental line is more highly methylated than the other parent, (P1 > P2 and P2 > P1), those for which both parents are methylated with equal levels (P1 = P2 > 0), and those in which no methylation is detected in either parent (P1 = P2 = 0).

In all four categories of each cytosine context, the discrepancies between hybrid and MPV methylation patterns in different parental combinations may be caused by the interaction between parental genomes. We were able to make the following generalizations following a paired-sample *t* test (Figure 2A,B and Appendix A): (i) the methylation levels of hybrids compared to MPV slightly increased in the unmethylated parental CCGG and CCWGG sites; (ii) the methylation levels of CCGG in hybrids were nearly the same as the MPV when parents were equally methylated; and (iii) if a hybrid differed significantly from MPV in methylation levels, then there were also methylation differences between reciprocal hybrids in the same cross combination. Moreover, we also found that apart from methylation status of P1 > P3, the hybrid with the hyper-methylated parent as maternal parent was more highly methylated than the other hybrid with the same parent as paternal parent when parents were differentially methylated in terms of average methylation levels (Figure 2A,B). These results indicated that there were differences in methylation patterns between reciprocal hybrids and that these differences may have resulted from parent-of-origin and cytoplasmic effects.

To ascertain whether the same genomic regions were targeted for change of the methylation status in reciprocal hybrids compared to their parents, an intersection analysis was conducted. This analysis was applied to differential methylation sites (DMSs) as described in the Materials and Methods section. Results showed that from 14.88 to 30.45% of methylation sites were shared by parental differentially methylated sites (PDS) and reciprocal hybrids differentially methylated sites (HDS) (Figure 2C). Subsequently, we searched for those sites that simultaneously inherited maternal line methylation status or simultaneously inherited paternal line methylation status for the reciprocal hybrid (called parental selective difference sites, PSD) (Appendix A). Among the sites of intersection between PDS and HDS (Ins), the results showed that PSD sites accounted for 74.30–88.93% of Ins and accounted for 13.23–22.63% of HDS. In addition, the number of sites that inherited maternal methylation status was the majority (Figure 2C,D). Overall, these results indicated that inheritance of parental methylation in hybrids was influenced by parent-of-origin and cytoplasmic effects.

### 2.3. The Non-Additive DMSs Are Predominantly Hypo-Methylated Sites

Cytosine methylation levels in an F_1_ hybrid may be additive or non-additive relative to its parents. We considered a DMS to be an additive site when methylation level of hybrid was equal to MPV, and conversely, a DMS was considered non-additive when methylation of the site was unequal to MPV.

Although hybrids were similar to MPV in overall methylation levels, there were still a number of DMSs identified in comparison of F_1_ hybrids and MPV in immature seed. In CCGG sites, F32 had the greatest number of non-additive DMSs (3635), while other hybrids had around 1500 DMSs. Similarly, in CCWGG sites, F32 also had the highest number of non-additive DMSs (4328), and the number of DMSs of other hybrids ranged from 2059 to 2746 (Figure 3A). In addition, regardless of cytosine context, most non-additive DMSs in all hybrids were parentally equally methylated (Figure 3B), which indicated that non-allelic epigenetic interaction in seed significantly contributes to heterosis. To further investigate the effect of cross parental methylated differences on the non-additive DMSs, those DMSs were divided into the following six categories according to the definition of the Materials and Methods section, “Below” or “Above” (a hybrid has lower or higher methylation levels than both parents when parents were equally methylated), “LP” or “HP” (the methylation levels of hybrids were similar to those of lowest or highest parents when parents were unequally methylated), “<LP” or “>HP” (the methylation levels of hybrids were transcendent to those of lowest or highest parents when parents were unequally methylated). Results showed that the top three patterns of non-additive DMSs in all hybrids were “Below” (46.73–69.84%), “Above” (16.51–35.39%) and “LP” (7.14–20.07%) irrespective of the cytosine context. Meanwhile, the vast proportion of DMSs were hypo-methylated sites compared to MPV (Figure 3C,D). Furthermore, we mapped non-additive DMSs to the genome and found an increased proportion in the genic regions, particularly in the region of the gene body at two methylation contexts (Figure 3E,F). These results suggested that methylation level of both reciprocal hybrids was lower than the lowest parent in the majority of the non-additive sites, especially in the genetic region, and the over-dominant demethylation in hybrids happens at two methylation sites.

Due to the presence of DMSs in reciprocal hybrids, a non-additive DMS in one hybrid does not necessarily occur in the other hybrid. We searched for DMSs common to the reciprocal hybrids among these non-additive DMSs and classified them by the same method as non-additive DMSs. Analysis showed that reciprocal non-additive DMSs and non-additive DMS shared the same general trends in any aspect, but with three differences: (i) although most reciprocal non-additive DMSs were located in the parental-equal methylated sites (from 63.60 to 87.72% at two cytosine context), the proportion of those DMSs in the sites with parental methylation differences increased; (ii) compared to non-additive DMSs of one hybrid, more reciprocal non-additive DMSs occurred in genic regions (48.21–63.37% in CCGG context and 29.28–34.49% in CCWGG context), especially in the CCGG sites, with the F1221 combination accounting for more than 60% of the total; and (iii) “Below” remained the dominant pattern of DMSs (37.31–56.80% in CCGG sites and 56.80–76.11% in CCWGG sites), but the proportion of some combinations decreased slightly (Figure 4A,B). The data suggested that reciprocal non-additive DMSs in any combination undergo substantial over-dominant demethylation at both contexts of methylation sites and is mostly influenced by non-allelic interactional effects, while this phenomenon occurs mainly in gene regions implying that its influence of gene expression results in the phenotypic heterosis.

### 2.4. Genes Involved in Various Biological Processes Show Non-Additive Methylation in Hybrid Seed

Changes in DNA methylation status generally affect gene expressions and genes showing non-additive methylation in hybrids possibly cause heterosis [13,15,30]. Hence, non-additive methylated genes (DMGs) in hybrids were analyzed. As described in the Materials and Methods section, only reciprocal non-additive DMSs in both paired hybrids were mapped to gene regions (2 kb of gene upstream or downstream regions and gene bodies) and DMGs were obtained for all cross combinations.

In CCGG sites, the F1221, F1331 and F2332 combination identified 240, 175 and 242 DMGs, respectively, and 149,113 and 152 genes in three combinations, respectively, were identified in CCWGG sites (Figure 4C,D). In sum, 556 and 363 non-redundant genes in CCGG or CCWGG showed non-additive methylation in all cross combinations, and only three genes are commonly found in CCGG and CCWGG context (Figure 3C,D). Moreover, a few genes had multiple non-additive DMSs assigned and these showed different, non-additive patterns among DMSs (Appendix A).

Finally, we performed functional annotation (Appendix A) and enrichment analysis of these DMGs. Although there was no shared significantly enriched GO (Gene Ontology) term in all three combinations (Fisher’s text, *p* < 0.05, at least five annotated genes were kept), some common GO terms were enriched at lower significance (Appendix A). These GO terms of CCGG sites were involved in multiple biological processes, such as energy and substance metabolism, gene transcriptional regulation, cellular processes, immune response and growth and development. Similarly, enriched GO terms of DMGs in CCWGG sites were also involved in various biological processes including positive regulation of nucleic biosynthetic and metabolic process, positive regulation of nitrogen compound metabolic process, positive regulation of transcription and DNA-templated. The above results were found for all types of organ hybrids in different crops, such as soybean flowers [25], rice panicles [31] and pigeonpea seedlings [12]. The remodeling of DNA methylation in the gene region may be interlinked with heterosis by affecting the expression of specific genes for various biological processes, and if this remodeling forms immature seed and can be transmitted with cell division to a certain period, then DNA methylation modification in hybrid contemporary seeds is associated with the initiation of hybrid advantage.

### 2.5. Non-Additive Methylation of Transcription Factors and Hormone-Related Genes in Hybrids Seed

As transcriptional regulation biological processes were enriched in DMGs of reciprocal hybrids seed, we used PlantTFDB [32] to search for transcription factors among DMGs and identified 38 transcription factors. These genes belong to different families and most showed hypo-methylated patterns (“LP”, “<LP” and “Below”) in hybrids (Appendix A). More interestingly, the annotation of those DMGs showed that nearly half of them were involved in phytohormone-related biological processes, especially abscisic acid pathways. Furthermore, some of the factors were associated with photosynthesis, flowering and photoperiod response, defense response. These results imply that hypo-methylation of transcription factors might cause an expression change in reciprocal F_1_ hybrids. This leads to changes in regulation of hormone biosynthesis or hormone signaling pathways, as well as light-related biological processes and defense response, resulting in phenotypic differences between hybrids and their parents.

Hormone-related biological processes are integral to hybrid vigor in plants [7,33,34]. Although the DMGs in this study were not enriched in the hormone-related GO term, the majority of transcription factors were associated with hormones suggesting that hormone-related pathways play a potential role in the initiation of soybean heterosis. Analysis revealed that a total of 78 genes involved in hormone biosynthesis or signaling pathways are non-additively methylated in reciprocal hybrids, and 10 genes were shared in multiple cross combinations, one (*Glyma.09G084300*) of which appeared in all combinations and responses to auxin stimulus (Appendix A). Meanwhile, the three genes (*Glyma.04G116700* in F2332 and *Glyma.12G124400*, *Glyma.12G124500* in F1221) are shared in different methylation contexts of one combination; interestingly these genes were also associated with auxin response. Remaining genes being those related to polar transport, homeostasis, biosynthesis and signaling of auxin; biosynthesis, metabolic regulation and signaling of jasmonic acid and salicylic acid; biosynthesis and signaling of brassinosteroid and ethylene; signaling of cytokinin, gibberellin and karrikin; transport and signaling of abscisic acid; as well as polyamine catabolic process. As with non-additive methylation of transcription factors, the majority of these phytohormonal genes show “LP” and “Below” patterns (including the four genes that are shared in all combinations or all methylated types of one combination), suggesting that multiple hormone-related pathways are hypo-methylated in hybrid seeds. Hence, we conclude that phytohormone signaling pathways may also play a role in the formation of heterosis in soybean.

## 3. Discussion

Crossing parents of different genotypes results in epigenetic reprogramming of the hybrid progeny creating differences in methylation between hybrids and their parents, as already reported in several plant species [10,11,12,13,35]. Studies on the flag leaf of soybean during the grain filling stage showed that four hybrids had lower methylation levels at CCGG sites as compared to either parent [27]. Using soybean leaves of 15-day-old seedlings showed that total relative DNA methylation level of reciprocal hybrids was lower than the corresponding MPV [36]. In our study, DNA methylation similarly undergoes rearrangement in immature soybean seeds (Figure 1 and Appendix A). The overall methylation levels of immature seeds of major hybrids were equal to or slightly lower than MPV, but this difference was not statistically significant, meanwhile the global methylation level of F32 at the CCWGG sites was significantly higher than that of the two parents, which may result from interactional effects of specific parental species (Appendix A). These phenomena implied that reprogramming the DNA methylation profiles of hybrids differs across species, developmental periods and cytosine methylation contexts, and that DNA methylation, as gene expression, is a spatiotemporal product of specific developmental stages [37]. Although the total level of methylation did not vary significantly between hybrids and corresponding MPV, many DMSs were still detected (Figure 3), indicating that the presentation of hybrid vigor may not require comprehensive remolding in DNA methylation of the whole genome for methylation contexts of our analysis, but only changes in DNA methylation at a range of key sites [15]. It has also been demonstrated in studies on fruit numbers of tomato and biomass of *Arabidopsis* that heterosis can be controlled by a single gene [5,6,7] or influenced by a set of key genes [38]. In addition, DNA methylation and the majority of hyper-methylated sites were found predominantly in transposable elements and intergenic regions (Figure 1 and Appendix A). A stable repressive epigenetic mechanism in these regions maintains genome stability by suppressing their activity [39,40]. We also found that CCGG methylation has a higher total level than the CCWGG and that their distribution across the genome differs, especially in the region of the gene body, where the CCGG sites are more numerous than the CCWGG sites and have a higher methylation level than the upstream and downstream 2000 bp (Figure 1 and Appendix A), suggesting that unlike methylation context, they may play different roles in regulating gene activity [6,15].

The two classic hypotheses for heterosis, dominance and overdominance, are both based on differences between the parental genomes [3,4]. Nonetheless, the epigenome altering of hybrids at the molecular aspect does not always occur in regions or loci that differ between crossing parents. For example, a large number of non-additive-expressed sRNA clusters of *Arabidopsis* hybrid seedlings were located in the parental non-differential region [13], and contemporary rice hybrid seeds exhibit non-additive methylation region predominantly present at the parent-equal region [16]. In the present research, many non-additive DMSs were also derived from the same methylated region of both parents in soybean (Figure 3 and Figure 4), suggesting that the interaction of non-allelic loci mainly affects altered methylation of hybrid contemporary seeds, but not the interactional effect of allelic loci. Another discovery is that some DMSs were present in the reciprocal hybrids for all combinations, a proportion of these DMSs were related to parent-unequal methylation region (Figure 2). Further analysis found that 74.30–88.93% of those DMSs in this region belong to parentally biased heredity of DNA methylation status. These parent-of-origin and cytoplasmic effects are also seen in other plants [41,42,43], which are associated with phenotypic dominance of hybrids [6,44].

Whether DNA methylation modification of contemporary hybrid affects the initiation of heterosis is a question worth discussing. In *Arabidopsis*, the active *FLC* chromatin state that is established in developing embryo can be maintained throughout the seed formation stages leading to active gene expression, suggesting that the epigenetic memory can be transmitted across multiple growth periods through cell division to influence gene expression at some special developmental stage [23]. Meanwhile, two studies have revealed significant DNA demethylation in hybrid contemporary seed embryos [16,43]; these hypo-methylations may activate long-term expression of genes or influence DNA methylation during subsequent developmental periods [16]. In addition, a study of dynamic changes of DNA methylation during soybean seed development shed light on that CHH methylation increased significantly with developing, while CG and CHG remained largely unchanged, and the differentially methylated regions at different periods for CHH sites were concentrated in the transcribed genes [20]. Another study on the dynamics of DNA methylation in soybean from seed formational periods to early seedling stages showed that CG and CHG methylation patterns remain unaltered throughout developmental periods and that the number of hyper-methylated sites in the CHH context increases as seeds mature, with the function presumably to inhibit TE activity. Moreover, this study also confirmed that the expression of important genes for seed formation was not disturbed by DNA methylation, and loss of non-CG methylation does not affect normal seed development [21]. Our results showed that most non-additive DMSs are hypo-methylation sites (Figure 3 and Figure 4). Interestingly, these DMSs were more concentrated than the natural distribution of methylated sites in the gene region containing the gene body and its upstream and downstream 2000 bp, especially in the CCGG context where the proportion of DMSs in the gene region can exceed 50% in some combinations (Figure 3 and Figure 4). Taking the above results together, we hypothesize that since the methylation profiling developed during soybean seed development is not exclusively associated with embryonic development, it must perform some function at one or more specific periods during the subsequent developmental process. It can be further concluded that some non-additive DMSs do not perform their functions during all the whole embryonic stages, but these DMSs are delivered to certain specific stage tissue via cytokinesis and may activate heterosis-related gene expression. Thus, DNA methylation in immature contemporary seeds of hybrids plays an important role in the initiation of hybrid vigor.

DNA methylation may contribute directly to heterosis by affecting gene expression in the methylated region, with CHH methylation altering circadian rhythm-related genes expression leading to an increase in *Arabidopsis* hybrid biomass [6], and CHG methylation controlling some genes in the young rice panicles where their expression levels are biased towards one parent and ultimately lead to increasing yield [15]. Our enrichment analysis of three combinatorial reciprocal non-additive DMGs results illustrated many biological processes related to gene regulation, and some DMGs are transcription factors that directly regulate the expression of a series of downstream genes (Figure 4 and Appendix A). Further analysis of these transcription factors revealed that most genes were involved in hormone metabolism and signaling pathways (Appendix A). Among these, several genes of these were shared for different combinations related to auxin. Moreover, 78 DMGs related to hormone biosynthesis or signaling pathways in all combinations were detected, suggesting an important role for hormones in heterosis (Appendix A). In fact, many previous studies have shown that phytohormones can influence the formation of hybrid vigor, such as auxin and ethylene, which contributes to early biomass advantage in *Arabidopsis* hybrids [7,45]. Additionally, most of hormone-associated DMGs show “Low/below” non-additive pattern in our analysis suggesting that these genes are hypo-methylated in hybrid seeds and might be activated in soybean embryos (Appendix A). A trade-off between growth and defense hypothesis used to explain growth and biomass heterosis in plant hybrids, which refers to the fact that under normal growth conditions without external stress, the basal defense genes in hybrids are maintained at low parental levels to preferentially supply the products of material and energy metabolism for growth and thus the growth of the hybrid is superior to that of the parents [1]. Interestingly, related-defense hormone pathways, such as salicylic acid and jasmonic acid, also appear in DMGs, that mostly show the “Above” pattern and defensive biological processes were also enriched in reciprocal hybrids (Appendix A), implying that DNA methylation may also regulate the stress response systems of hybrids to participate in a trade-off between growth and defense.

Non-additive alterations in epigenetic modification of hybrids may lead to non-additive expression of genes [9], but we were unable to find the common gene between the non-additive methylated genes in our study and the non-additive expressed genes in previous studies [26,27] due to differences in varieties and planting environments. However, several related-yield genes or photosynthesis were still sought out in reciprocal non-additive DMGs. The details are as follows: *Glyma.08G183500* [46], *Glyma.19G009900* [47], *Glyma.19G067900* [48] and *Glyma.07G140200* [49] relates to seed yields, and overexpression lines of *Glyma.08G183500* can lead to the increase of heavy 100-seed weight in soybean; the function of flowering factors GmFT2a and GmFT5a are affected by *Glyma.19G122800* [50] and *Glyma.19G224200* [51,52]; *Glyma.03G137000* encode chlorophyll biosynthesis of soybean to directly control photosynthesis [53]. These DMGs also corroborate the potential role of DNA methylation modification in heterosis initiation of soybean hybrid contemporary seeds, but whether these genes are actually involved in trait dominance of hybrids needs further verification.

## 4. Materials and Methods

### 4.1. Plant Material

The F1 hybrids were generated by artificial crossing in the experimental field of Jilin Agricultural University (Changchun City, Jilin Province, China) according to Griffing’s diallel cross design for method I (Table 1), Jilin38 (code as P1), Y3 (code as P2) and Jilin47 (code as P3) as hybrid parents. These parents, with the exception of P2, are the main cultivars in Jilin Province. Furthermore, previous studies have shown a significant heterosis in yield traits in hybrids mated with them as parents [28,36]. After crossing, the immature seeds were sampled from R6 (full seed) stage, frozen immediately in liquid nitrogen and stored at −80 °C until use.

### 4.2. DNA Extraction and Whole Genome Methylation Sequencing

Total genomic DNA was isolated from all samples with modified CTAB method [54]. Then the integrity and purification of DNA were tested by agarose gel electrophoresis and spectrometric measurement. The genome methylation status of soybean seeds was detected based on MethylRAD technology (Shanghai OE Biotechl Co., Ltd., Shanghai, China). The methylation library construction and high-quality methylation sequencing data acquisition were performed by Song et al. [28]. After obtaining the chromosome position of all sites was aligned to each different functional element of the genome (such as gene body, exon, intron, transposon and intergenic regions) using custom Python scripts. In addition, the reads per million (RPM) were used to calculate DNA methylation levels of each restriction site (CCGG/CCWGG) for determining the relative quantification of MethylRAD data, and the methylation levels of each element are equal to the sum of the methylation levels of sites in the element.

### 4.3. Differentially Methylation Analysis

Comparison of methylation differences between any two samples and genomic regions was performed by *t* test and significant differences were considered if *p* < 0.05; methylation level of site in the genomic region was also counted as the average of the methylation levels at this site for all samples. Differential methylation sites (DMSs) between two samples were identified using the R package edgeR [55]. A site with log2|FC| >1 and *p* < 0.05 found by edgeR were assigned as differentially methylated. Categorizing sites into methylation patterns, for non-additive remodeling pattern in hybrids, was performed by comparing the methylation level in F_1_ with the mid-parent values (MPV), also using edgeR, and a site with log2|FC| > 1 and *p* < 0.05 were considered to be non-additively methylated; in contrast, sites that were differentially methylated in the MPV with *p* > 0.05 were considered to be additively inherited patterns. Non-additively methylated DMSs are further subdivided into four patterns when methylation level of parents is different, “HP” or “LP” (methylation level of a hybrid is similar to the highest or lowest methylation level of parent), and “>HP” or “<LP” (methylation level of a hybrid is above or below that of both parent). When the methylated site of parents are equally methylated, non-additively methylated DMSs are further classified into patterns “Above” or “Below” (methylation level of a hybrid is above or below that of both parents).

The methods of DMSs treated as reciprocal non-additive DMSs and defined non-additively methylated genes between reciprocal hybrids and MPV are referenced to Zhou et al. [16]. DMSs will be considered as reciprocal non-additive DMSs when methylation change in both reciprocal hybrids show identical (such as both “HP” in F12 and F21) or similar (such as “HP” in F12 and “>HP” in F21) methylation altered trends. These reciprocal non-additive DMSs will be assigned to individual genes, only DMSs located within 2 kb of gene upstream or downstream regions and gene bodies are assigned to the nearby genes. Genes with multiple assigned DMSs are treated as one non-additively methylated gene no matter whether the non-additive status of each DMS is concordant or conflicted.

### 4.4. Bioinformatic Analysis

The soybean reference genome (Wm82.a2.v1) was used as reference genome in this study. Transposon elements (TE) annotations from SoyTEdb [56] based on Wm82.a1.v1 genome were converted to the Wm82.a2.v1 genome by Blast+ [57], and the results used to further analysis. All differentially methylated genes (DMGs) were annotated using SoyBase database (https://www.soybase.org/ accessed on 15 March 2020) and TBtools [58] (Fisher’s exact test, *p* < 0.05, at least 5 annotated genes were kept) was used to find significantly enriched GO terms in the input list of DMGs.

## 5. Conclusions

Our study confirms that in contemporary immature hybrid seeds of soybean, DNA methylation profiles are methylated or demethylated at a range of CCGG and CCWGG sites compared to their parents, rather than global alteration of the whole genome. At the same time, maintenance methylation status of one parent occurs at the reciprocal hybrid’s differential sites result from the parent-of-origin and cytoplasmic effects in parent-unequal methylated regions. Most of the non-additive DMSs we retrieved in the hybrids were hypo-methylated sites compared to their corresponding MPV, meanwhile these sites were more concentrated in gene regions than the natural distribution of DNA methylation sites, illustrating that they are associated with gene expression. In addition, the enrichment results for reciprocal non-additive DMGs were concentrated in biological processes, such as gene expression regulation, cell growth processes and stress response, some of which are transcription factors or involved in hormone synthesis and signaling pathways. These results are consistent with previous studies and suggest that DNA methylation in hybrid contemporary seeds play a role in the initiation of heterosis.

## Figures and Tables

**Figure 1 plants-11-01136-f001:**
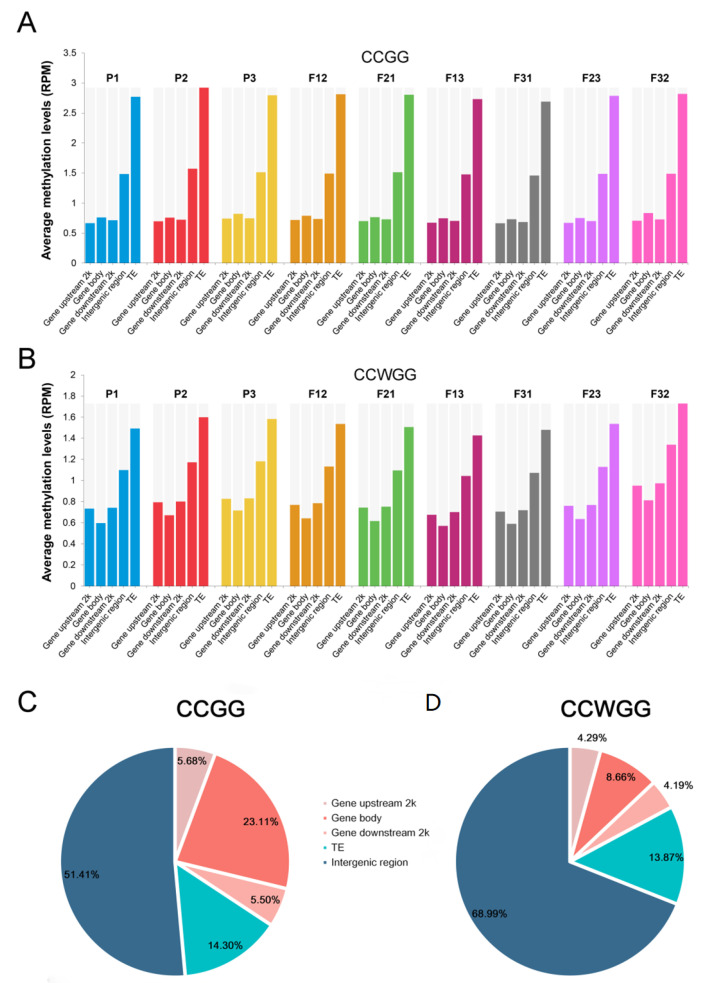
DNA methylation landscapes of hybrids and their parental lines in hybrid contemporary seeds. (**A**,**B**) Relative average methylation levels in different genomic elements. (**C**) Distribution of methylation sites in soybean genome. (**D**) The significant difference of CCGG and CCWGG methylation levels. The number above black line of two sites is the *p* value determined by the paired Student’s *t*-test.

**Figure 2 plants-11-01136-f002:**
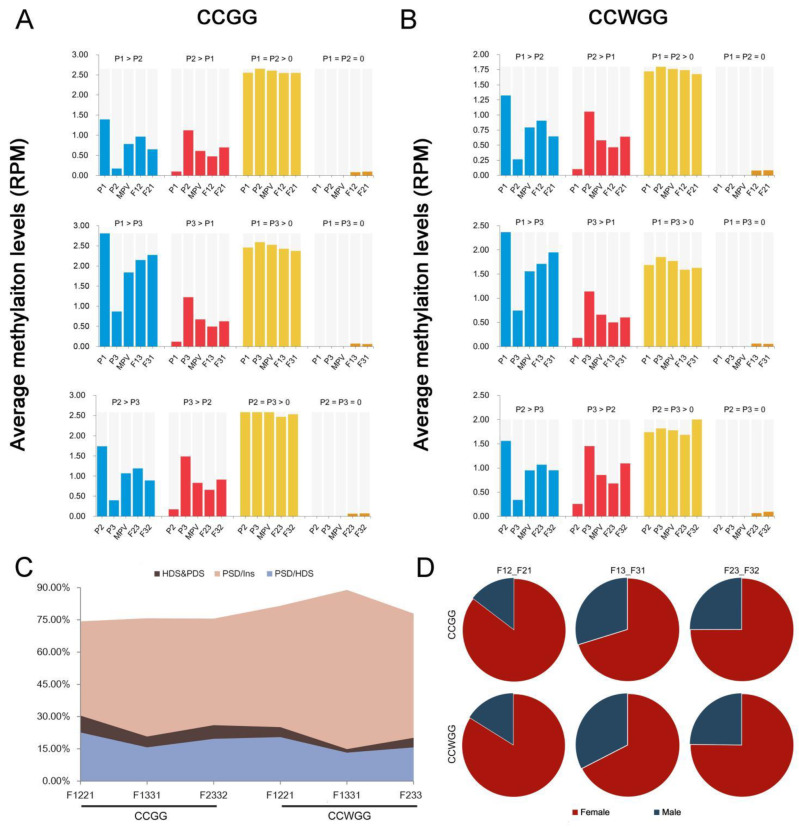
Association of DNA methylation in reciprocal hybrids with parental methylation status. (**A**,**B**) Average methylation levels at three combinations in developing seeds. Four groups are classified based on parental methylation differences in two cytosine contexts, including one parental line higher methylated than another parental line (e.g., P1 > P2 or P2 > P1); both parents are methylated with equal methylation levels (e.g., P1 = P2 > 0); cytosine contexts showing no methylation in two parents (e.g., P1 = P2 = 0). (**C**) Association of DNA methylation in reciprocal hybrids with parental-unequal methylated sites. HDS, reciprocal hybrids differentially methylated sites; PDS, parental differentially methylated sites; HDS&PDS and Ins, intersection ration of HDS and PDS; PSD, parental selective difference sites; PSD/Ins, PSD as a percentage of Ins; PSD/HDS, PSD as a percentage of HDS; F1221/F1331/F2332, the hybrid combination, e.g., F1221 is a combination using P1 and P2 as crossing parents, the same as figure below. (**D**) Parental selective difference sites in reciprocal hybrid’s unequal region. F12_F21/F13_F31/F23_F32, the methylated differential site of reciprocal hybrids, e.g., F12_F21 denotes those sites between F12 and F21.

**Figure 3 plants-11-01136-f003:**
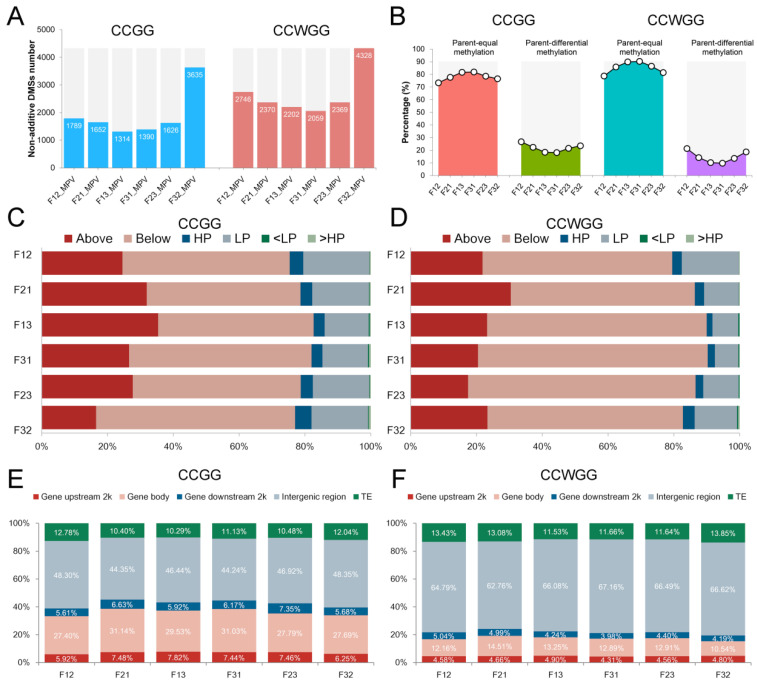
Non-additively methylated sites in all hybrids. (**A**) The non-additive methylation sites number of all hybrids. (**B**) Association of non-additive methylation sites with parental methylation status. (**C**,**D**) Non-additive categories of methylated sites in all hybrids. These sites were classified when methylation change in all hybrids is identical. “Above” or “Below” means methylation level of one hybrid is greater than two parents or less than two parents of parental equivalent regions. In parental unequal regions, “HP” or “LP” means methylation level of one hybrid is similar to the highest or lowest methylation level, “<LP” or “>HP” means methylation level of one hybrid is lower than in lowest parent or higher than in highest parent. (**E**,**F**) The distribution ratio of non-additive sites in three reciprocal hybrids for two methylation contexts.

**Figure 4 plants-11-01136-f004:**
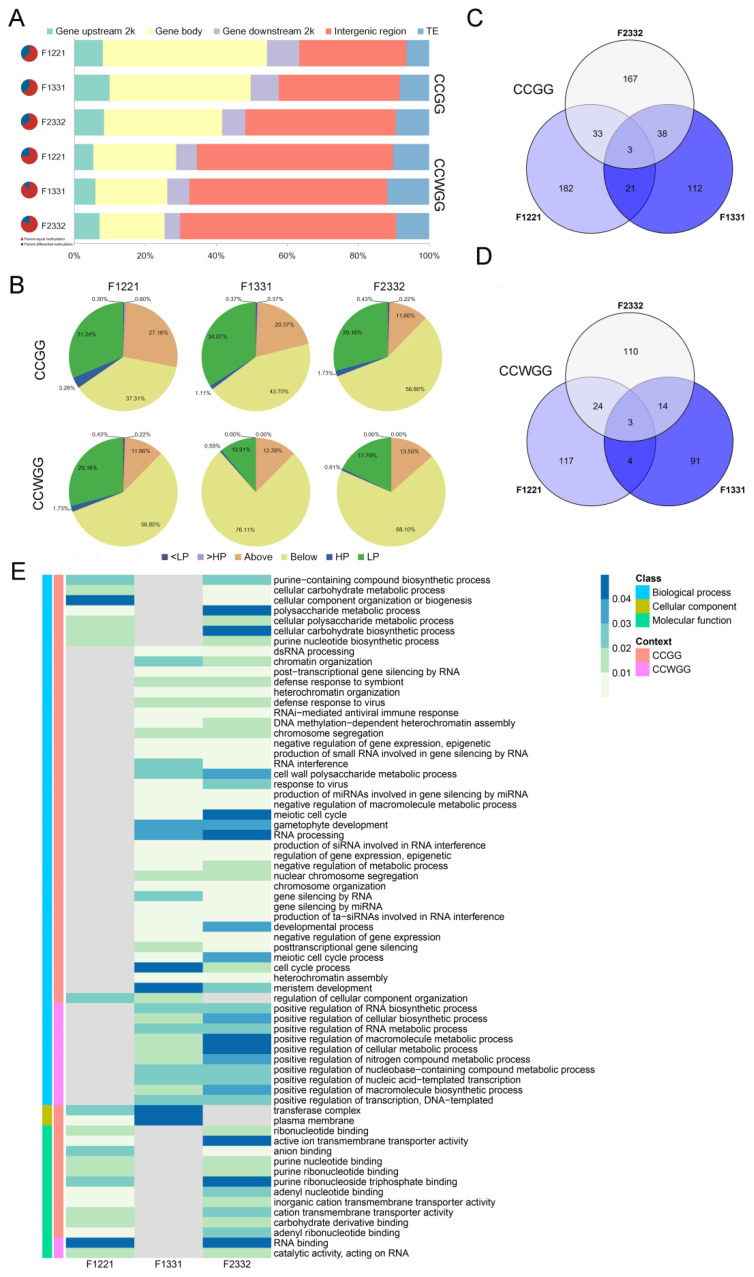
Non-additively methylated sites in reciprocal hybrid seeds. (**A**) The proportion of reciprocal non-additive methylated sites (re-DMSs) in whole genome and parent methylated region. The left pie chart shows the ratio of these sites in parent methylated region, red color denotes parent-equal regions and blue color symbolizes parent-unequal regions. The right chart displays the scale of these sites in genomic element. (**B**) Non-additive categories of re-DMSs in three group reciprocal hybrids. Re-DMSs were classified when methylation change in both reciprocal hybrids is identical (e.g., both “<LP” in F12 and F21). “Above” or “Below” means methylation level of reciprocal hybrids is greater than two parents or less than two parents of parental equivalent regions. In parental unequal regions, “HP” or “LP” means methylation level of reciprocal hybrids is similar to highest or lowest methylation level, “<LP” or “>HP” means methylation level of reciprocal hybrids is lower than in lowest parent or higher than in highest parent. (**C**,**D**) Statistics of re-DMSs in three reciprocal hybrids for two methylation contexts. (**E**) Enriched GO processes of non-additively methylated genes for three reciprocal hybrids. The color is bluer as *p* value is more insignificant, grey means insignificant enrichment.

**Table 1 plants-11-01136-t001:** Crosses between parents, Jilin 38/Y3/Jilin47 and respective F1 hybrids.

Parents	Jilin 38 (P1)	Y3 (P2)	Jilin 47 (P3)
Jilin 38 (P1)	-	P1 × P2	P1 × P3
Y3 (P2)	P2 × P1	-	P2 × P3
Jilin 47 (P3)	P3 × P1	P3 × P2	-

## Data Availability

All data generated or analyzed during this study are included in this article and its Appendix A.

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
