# Peer review of "Heterosis and Differential DNA Methylation in Soybean Hybrids and Their Parental Lines"

_plants, 2022, doi:10.3390/plants11091136_

Round 1

Reviewer 1 Report

The paper “Heterosis and differential DNA methylation in soybean hybrids and their parental lines” by Liangyu Chen et al reported the analysis of DNA methylation patterns of three combinations of reciprocal hybrids and the parents by MethylIRAD technology on CCGG/CCWGG sites.

The results highlighted a DNA methylation reprogramming in hybrids, involving genes associated with hormone biosynthesis and signalling, defence response, photosynthesis, flowering and photoperiod response, suggesting a role for DNA methylation in the initiation of heterosis.

The results are interesting but to support the conclusions, the expression analysis on some key genes differentially expressed should be performed.

Furthermore, the paper must be carefully reviewed by a native English speaker, in many points it’s not clear and typos are present.

Caption Fig 1,2,3,4: (A) to (B)….. (E) to (F) Should be (A) and (B), (E) and (F).

Fig 1,2,3,4: the graphs are small and the labels are very difficult to read.

In my opinion the paper in the present form is not suitable for publication in “Plants”.

Reviewer 2 Report

The manuscript submitted for review meets all the requirements of the manuscripts in PLANTS.  The topic of methylation, heterosis and epigenetics taken up by the authors is the current research task and is also consistent with the thematic profile of the journal. 

I have no major comments on the manuscript. The introduction contains all the necessary information to introduce the reader to the topic. 

There is probably a typo in the first sentence of the "Plant material" chapter.

In this chapter I miss at least a brief description of plant growth. Did the plants after artificial crossing develop under controlled conditions in the laboratory (phytotron chamber, greenhouse) or under field conditions?

The results chapter includes easy-to-read pictures that have been extensively described. However, the sentence in line 257-259 is not very clear to me. I understand the general context, however, the authors should be clearer about their hypothesis. The authors applied their results very well to the available literature data. the discussion section is very well written. The authors argue the hypotheses very well. I have a question if in line 330 there should be genotypes instead of phenotypes. 

The literature cited by the authors of the manuscript is up-to-date and well-chosen.

The only thing that I missed at work was the more widely described practical aspect of the presented data.

As the authors argue, the methylation profile depends on the development stage of the plant, and hybrids of different plant cultivars show different methylation patterns.
For me, an interesting aspect of the work would be to extend the presented research to the analysis of the proteome and metabolome of hybrid plants (F1) and their parents (P). Such a comparative analysis would allow us to expand our knowledge of the exact mechanism of higher yielding and greater resistance of hybrids.

I am also curious why the authors decided to analyze the immature seeds. Unfortunately, I did not find the answer in manuscript why the authors analyzed this particular developmental stage of soybeans.

I am eager to read the next publications of these authors.

Summarize my review, I believe that the manuscript should be published in PLANTS with minor revision. 

Round 2

Reviewer 1 Report

The revised version of the paper “Heterosis and differential DNA methylation in soybean hybrids and their parental lines” by Liangyu Chen et al. has been deeply improved but there are still some typos (line 61: “shown” should be “showed”, line 68: “probable causing” should be “probably causing” etc….

I suggest a careful revision by a native English speaker.

Author Response

Thanks
